# Postnatal Leptin Levels Correlate with Breast Milk Leptin Content in Infants Born before 32 Weeks Gestation

**DOI:** 10.3390/nu14245224

**Published:** 2022-12-08

**Authors:** Trassanee Chatmethakul, Mendi L. Schmelzel, Karen J. Johnson, Jacky R. Walker, Donna A. Santillan, Tarah T. Colaizy, Robert D. Roghair

**Affiliations:** 1Stead Family Department of Pediatrics, Carver College of Medicine, University of Iowa, Iowa City, IA 52242, USA; 2Department of Obstetrics and Gynecology, University of Iowa, Iowa City, IA 52242, USA

**Keywords:** enteral, infant, Holder pasteurization, leptin, premature, obesity

## Abstract

Perinatal leptin deficiency and reduced intake of mother’s milk may contribute to the development of childhood obesity. Preterm infants have reduced leptin production, and they are at heightened risk of neonatal leptin deficiency. Because fresh human milk contains significantly more leptin than donor milk, we used a cross-over design to determine if blood leptin levels in maternal milk-fed preterm infants fall during conversion to donor human milk. Infants born between 22 0/7 and 31 6/7 weeks gestation on exclusive maternal milk feedings were enrolled into a 21-day cross-over trial. On days 1–7 and 15–21, infants were fed maternal milk, and on days 8–14, infants were fed donor milk. On day 1, study infants had a mean postmenstrual age of 33 weeks. Plasma leptin correlated with milk leptin, and leptin levels in maternal milk far exceed the leptin levels of donor milk. Plasma leptin did not increase during donor milk administration, but it did following resumption of maternal milk (*p* < 0.05). In this crossover trial, preterm infant blood leptin levels correlated with milk leptin content. This suggests that preterm infants can enterally absorb leptin from human milk, and leptin-rich breast milk may be a targeted therapy for the prevention of obesity.

## 1. Introduction

Monogenic leptin deficiency leads to early childhood obesity [1]. Given leptin’s key signaling roles in the regulation of metabolism and energy homeostasis [2], children with leptin gene mutations and other forms of congenital leptin deficiency develop severe hyperphagia, reduced energy expenditure and obesity [3]. In fetal life, leptin is also an integral stimulus for hypothalamic development [4], and environmentally induced perinatal leptin deficiency has been implicated in the programming of childhood obesity as a consequence of acquired hypothalamic leptin resistance [2,5].

As an adipokine with a critical role in human neurodevelopment, there are redundant sources of leptin for the developing fetus, including maternal plasma, the placenta, and fetal adipocytes. Consistent with the timing of significant fetal fat mass accumulation, fetal blood leptin levels increase rapidly during the third trimester [6]. Because premature infants face the challenge to grow and develop in the presence of profound leptin deficiency, as a consequence of a lack of maternal and placental leptin sources and a limited amount of accumulated adipose tissue [6], it is not surprising that premature infants are at increased risk for obesity and type 2 diabetes mellitus [7].

Since its discovery in 1994, leptin replacement therapy has been extensively investigated in rodent models as well as in children and adults, where parenteral leptin replacement therapy has been approved for the treatment of congenital leptin deficiency, diabetes, and obesity [8,9]. Upon replacement therapy with recombinant human leptin, individual patients achieve weight loss, a reduction in body fat mass, and a normalization of their metabolism [10]. Although orally ingested leptin has been shown to be absorbed by neonatal rats and adult humans [8,9], we are not aware of any studies that have evaluated leptin replacement in preterm populations.

The presence of significant amounts of leptin in breast milk suggests that breast milk could be a potential source of leptin in neonatal life [5]. Even so, it remains unclear to what extent enterally administered leptin reaches the premature infant circulation [11]. Evidence also remains limited regarding leptin levels in infants fed by mother’s own milk versus human donor milk which contains significantly less leptin [12]. To our knowledge, there is no published evidence on whether preterm infants’ plasma leptin levels are influenced by type of breast milk received. We hypothesized that crossover to low leptin-content donor breast milk versus higher leptin-content maternal breast milk will correlate with lower circulating leptin levels.

## 2. Materials and Methods

We performed a nonblinded cross-over trial in infants admitted at the Stead Family Children’s Hospital neonatal intensive care unit (NICU), a level IV NICU in Iowa City, Iowa. Inborn infants born between 22 0/7 weeks and 31 6/7 weeks gestation or out born infants born at 22 0/7–29 6/7 weeks gestation that were transferred to University of Iowa prior to 30 0/7-week postmenstrual age (PMA) were screened for eligibility. Infants became eligible when they surpassed 26 6/7 weeks PMA and were receiving only enteral feeding with maternal milk, having discontinued parenteral nutrition at least 4 days prior. Additionally, mothers of eligible infants were required to have adequate breast milk supply for the length of the study. Exclusion criteria included infants with confirmed chromosomal anomalies, cyanotic congenital heart disease or history of abdominal surgery. Informed consent was obtained from parents, and the institutional review board at the University of Iowa approved the study protocol (IRB number 201905861). The research committee of the Human Milk Banking Association also reviewed and approved the study protocol.

After informed consent was obtained, a batch of frozen expressed maternal milk was allocated for the study period, labeled for study use, and stored in NICU nutrition area. At clinician’s discretion, all infants’ feeding volumes and the caloric density of enteral feeds were adjusted by the medical team based on their own assessments. Frozen maternal breast milk and donor human milk were used for the study. Donor milk was obtained from the Mother’s Milk Bank of Iowa, per standard NICU practice. Thawing of frozen breast milk was performed with the Medela Waterless Milk Warmer™. Breast milk was fortified with Enfamil Human Milk Fortifier (HMF) and Enfamil Term Formula Concentrate (Mead Johnson Nutrition, Chicago, IL, USA). 24kcal/oz fortification was prepared by mixing 5 parts human milk with 1-part HMF, 27kcal/oz fortification was prepared by mixing 3 parts human milk with 1-part HMF, and 30kcal/oz fortification was prepared by mixing 3 parts human milk with 1-part HMF and 1-part concentrate. These are the standard fortification recipes in use in the unit for all human milk, and they were not study-specific.

There were 3 weeklong phases in this research protocol. During phase 1, infants continued to receive fortified maternal milk. Study feedings were prepared by study personnel, once daily, with a 24 h supply of fortified maternal milk delivered to the infant bedside for administration by nursing personnel. The frozen maternal milk was used in a chronological order of the pumped dates. After one week, phase 2 began and infants received full enteral feedings of fortified donor milk for 7 days. Donor milk was allocated for the study period from donor milk supplies present in the NICU, from pools containing only donors who have consented to research use of milk. All donor milk used in the study was administered less than 12 months after the date it was expressed by the donor, as per Human Milk Banking Association of North America Guidelines. One week later, phase 3 began as infants were returned to a fortified maternal milk diet with study feedings prepared as described in phase 1.

A sample of maternal milk or donor milk (according to study phase) was collected for leptin analysis daily during the study protocol, prior to addition of human milk fortifier. The milk samples were stored at −80 °C, then thawed prior to centrifugation at 15,000× *g* at 4 °C. Consistent with recommendations from prior investigations, the fat layer was discarded, and the skim milk was subjected to analysis [13,14]. Infant blood was collected from clinically obtained blood samples to determine pre-prandial plasma leptin levels on days 7, 14, and 21. All infant blood samples were collected early in the morning (0400–0600) per unit standard practice, within 30 min of a scheduled feeding. A total of 200 microliters of blood were collected in EDTA tubes and processed within 4 h. Plasma was stored at −80 °C. Fasting plasma leptin and milk leptin were measured by the Quantikine human leptin ELISA (R&D Systems, Minneapolis, MN, USA). All samples were run in duplicate. Each assay included 7 standards and a quality control to ensure lot-to-lot consistency.

Maternal and infant demographic and clinical data were retrospectively collected using electronic medical records (Epic, Verona, WI, USA). Maternal medical history, age, gravidity, parity, and mode of delivery were obtained from the medical record. Gestational ages, sex, daily weight during the research period, prescribed medications and supplements, and fortification levels of infants were obtained from infant medical records. Measurements of length and head circumference were performed by research personnel weekly during the study protocol with a non-stretchable tape measure. Z-scores in weight, length and head circumference were calculated using the Fenton 2013 growth calculator for preterm infants [15,16]. All data were recorded and stored in REDCap version 8.3.2 [Vanderbilt, TN, USA]. Statistical analysis was performed using SigmaPlot 14 (Systat Software Inc., San Jose, CA, USA). Paired t-test was used for comparison of breast milk leptin levels or infant plasma leptin levels. Simple linear regression was used to determine the relationship of infant plasma leptin and breast milk leptin concentration, breast milk volume consumed, the amount of leptin received in the diet, the infant’s current weight or the infant’s weight gain. Post hoc multiple linear regression was used to test the assumption that infant plasma leptin levels could be predicted by the infant’s current weight and the amount of leptin in their diet. Statistical significance was defined by *p* < 0.05.

## 3. Results

Of 206 infants screened on admission for enrollment, 131 were excluded (73 due to inadequate breast milk supply prior to enrolment, 30 due to language or social barriers, 15 due to history of prior abdominal surgery, and 13 due to clinical research pause during the COVID-19 pandemic), 16 died prior to possible enrolment, and 6 were transferred to local NICUs prior to enrollment. Parents of 53 infants were approached for consent and 42 of them declined. Of 11 infants that were enrolled, 8 successfully completed the study. One infant was withdrawn from the study due to institution-wide halting of all research procedures during the COVID-19 pandemic, one was withdrawn due to inadequate maternal milk supply on day 4 of the study, and one was excluded after enteral feedings were discontinued by the clinical team during the study.

Representative demographic data are provided in Table 1. Additionally, of the 8 mothers, 2 were primigravid, one had diabetes and one developed chorioamnionitis, but none of them had preeclampsia or other hypertensive disorders of pregnancy.

Infant growth and nutrition parameters are provided in Table 2. All infants were diagnosed with bronchopulmonary dysplasia, 4 received systemic steroids and 6 received chlorothiazide diuretic. Three infants underwent sepsis evaluation during the study period, but all blood cultures were negative, and no one was diagnosed with necrotizing enterocolitis. Four infants received blood transfusions during the study for anemia of prematurity according to the institution protocol. Each infant had a normal fasting glucose level (3.6 mmol/L to 5.6 mmol/L) while on full enteral feedings before study entry, and additional nutritional labs were not obtained during the study.

As shown in Figure 1, the mean leptin level in donor milk (3.8 pg/mL) was significantly lower than the leptin levels in maternal milk during the first or last week of the study (580 pg/mL and 577 pg/mL, respectively).

Infant plasma leptin levels did not increase while infants received donor milk (989 pg/mL on donor milk, from 1434 pg/mL on maternal milk), but they significantly increased to 1774 pg/mL following conversion back to maternal milk (Figure 2), representing a 48% increase from baseline by paired analysis (Figure 3).

Overall, plasma leptin levels significantly correlated with the leptin content of the breast milk the infants had received earlier that same day (Figure 4). By simple linear regression, there was no correlation between leptin levels and the volume of breast milk consumed (R = 0.04, *p* = 0.84) or the infant’s incremental weight gain (R = 0.37, *p* = 0.08).

By multiple linear regression, leptin levels were predicted (R = 0.75, *p* < 0.01) by the infant’s current weight (simple linear regression R = 0.60, *p* < 0.01) and the amount of leptin in their diet (simple regression R = 0.40, *p* = 0.05), with the latter parameter estimated by multiplying breast milk leptin concentration and the volume of breast milk consumed. Figure 5 demonstrates the changes in plasma leptin and body weight for individual infants.

## 4. Discussion

Premature infants face the challenge to grow and develop in the presence of profound leptin deficiency [4,6]. They are born before the third trimester leptin surge and suffer from a lack of placental and transplacental leptin delivery [4,6]. Our study strongly suggests that the composition of breast milk influences leptin levels with direct correlation seen between breast milk leptin content and plasma leptin levels, as well as increased leptin levels upon conversion from leptin-depleted donor milk to maternal milk that had consistently higher leptin content.

Classically, leptin is an adipokine predominantly produced by mature adipocytes with a key role in energy-regulation, and in adulthood, blood leptin levels correlate with body mass index [2]. Beyond its role in adulthood, leptin exerts early neurotrophic effects to set the stage for its later roles in modulating energy expenditure and sympathetic activation [4,17,18,19]. Leptin appears to be so key to early life development that the placenta itself is an important source of leptin for the developing fetus [20,21].

In an animal model of prematurity-related leptin deficiency, mice with neonatal leptin deficiency develop features of adult metabolic syndrome, but neonatal leptin supplementation can prevent the neuropsychiatric impairment and hypertensive response to psychological stress that otherwise follows neonatal leptin deficiency [22,23]. Further studies in neonatal rodents have shown that oral supplementation with leptin protects against the development of adult cardiometabolic disease [24,25]. Moreover, orally ingested leptin was shown to be absorbed by the immature rodent stomach and transferred to the circulation [9,26].

In the human fetus, adipose tissue development begins in 2nd trimester with most major areas deposited by 28 weeks gestation, but fat mass increases significantly during the final trimester [27]. During the first and second trimesters, the mother and the placenta dictate fetal plasma leptin levels and multiple studies have shown a positive correlation of maternal leptin levels with advancing gestational age; the fetus only becomes a main contributor of leptin levels in the third trimester [6,28]. However, transplacental leptin delivery remains significant contributor to fetal leptin level up to term gestation with several studies demonstrating an abrupt termination of the physiologic leptin surge following delivery of both term and preterm infants [29,30,31]. Furthermore, after the initial decline in leptin level following delivery, premature infants continue to have profound leptin deficiency throughout the critical period of development that encompasses their neonatal course [6,32,33].

The source of nutrition influences postnatal hormone levels. The identification of leptin in breast milk suggests a potential role in the regulation of infant development [5]. Parenteral nutrition, human milk fortifiers and infant formulas do not contain leptin [34,35]. Moreover, the American Academy of Pediatrics recommends that for preterm infants, human donor milk is preferred over formula when mother’s own milk is unavailable [36,37]. Despite several benefits of human donor milk, the pasteurization process inevitably affects the levels of several hormones found in human milk, and shown in our study, leptin is among the hormones reported to be the most affected by the pasteurization process [12]. Unfortunately, we were not able to quantify the breast milk or infant levels of other cytokines and growth factors, and that is an important consideration when considering the limitations of our study.

Other investigations have provided converging evidence that leptin may play a role in the favorable growth profiles seen in formerly breast-fed children. Savino and colleagues reported that breast fed term infants have higher leptin levels than formula fed babies even though breast fed babies had lower body mass indexes [38]. Moreover, several studies reported inverse associations between breast milk leptin level and infant weight gain suggesting a role of milk-borne leptin in the early regulation of body weight and the prevention of adult obesity [39,40,41,42]. It is well known that breast milk leptin content is directly related to maternal body mass index and maternal plasma leptin [12,41,42,43], suggesting an important role for breast feeding in women with increased adiposity to lessen the risk of obesity in their offspring. In addition to leptin concentration, the volume of maternal milk received also appears to directly contribute to increased leptin levels in preterm infants [44].

In summary, our cross-over trial has shown that preterm infant blood leptin levels correlate with breast milk leptin content, and conversion from leptin-deficient donor milk to higher leptin-content maternal milk is associated with increased infant leptin levels. These data suggest that preterm infants can absorb leptin from human milk, with increased oral ingestion resulting in increased blood levels. Future studies are needed to assess the potential of enteral leptin supplementation as a targeted strategy to decrease the risk of childhood obesity among preterm infants.

## Figures and Tables

**Figure 1 nutrients-14-05224-f001:**
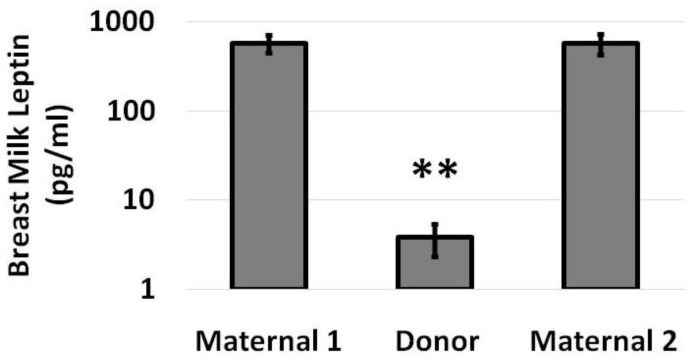
Breast milk leptin levels during the initial week of maternal milk, intervening week of donor milk and final week of maternal milk are presented on a logarithmic scale. Compared to leptin levels in donor milk, leptin levels in maternal milk were over 100-times higher. Data are presented as mean (SD) with N = 8 and ** *p* < 0.01 for Donor versus Maternal 1 and Donor versus Maternal 2.

**Figure 2 nutrients-14-05224-f002:**
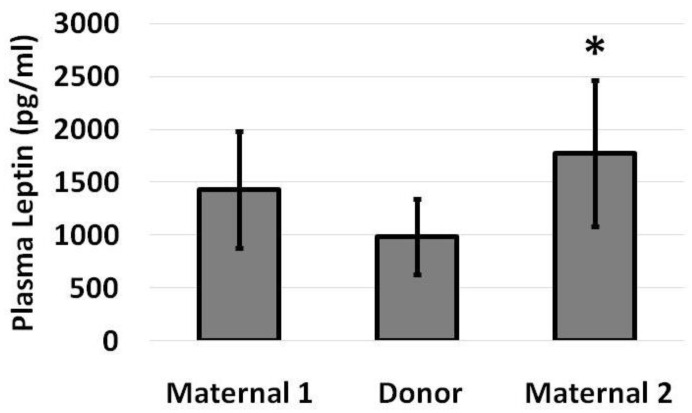
Plasma leptin levels among infants at the end of a week of maternal milk, an intervening week of donor milk and a final week of maternal milk. Leptin levels increased on conversion from donor milk back to maternal milk. Data are presented as mean (SD) with N = 8 and * *p* < 0.05 for Maternal 2 versus Donor.

**Figure 3 nutrients-14-05224-f003:**
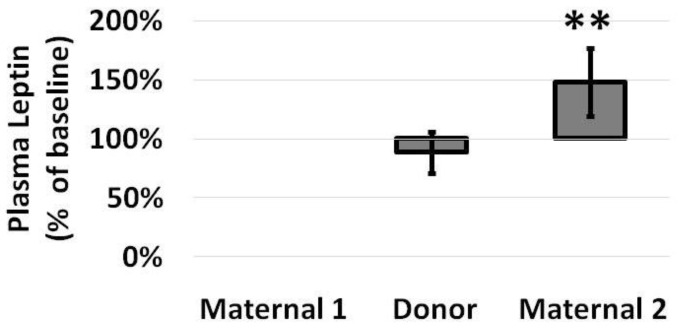
Plasma leptin levels, as a percentage of baseline values, among infants at the end of a week of maternal milk (all values 100% by definition), an intervening week of donor milk and a final week of maternal milk. Leptin levels only increased during the administration of maternal milk. Data are presented as mean (SD) with N = 8 and ** *p* < 0.01 for Maternal 2 versus Donor.

**Figure 4 nutrients-14-05224-f004:**
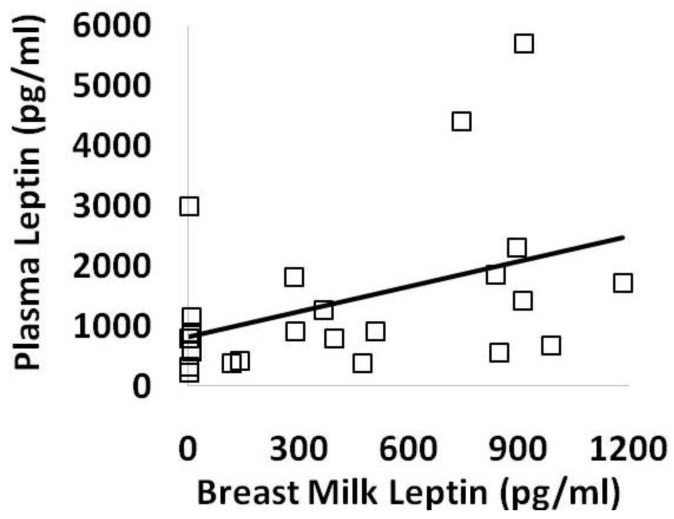
Plasma leptin levels were directly associated with breast milk leptin levels across the 3 weeks of the study protocol by simple linear regression (N = 24, R = 0.42, *p* < 0.05).

**Figure 5 nutrients-14-05224-f005:**
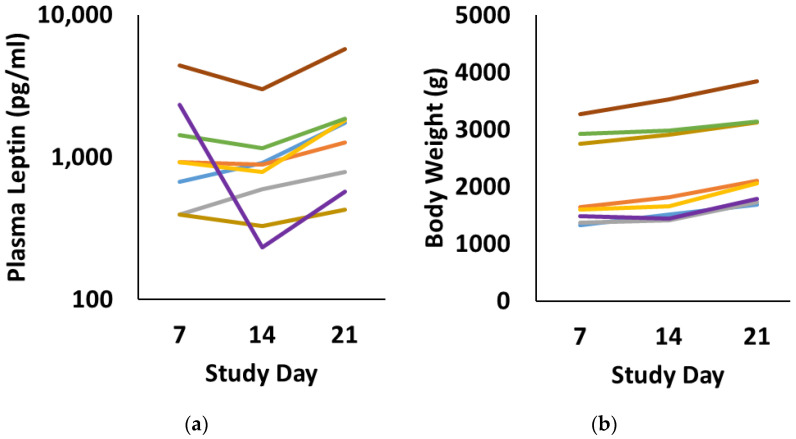
Longitudinal plasma leptin levels (**a**) and body weight (**b**) for each of the 8 infants that completed the clinical study. Study day 7 followed at least 7 days of maternal milk, study day 14 followed 7 intervening days of donor milk, and study day 21 was 7 days following a return to maternal milk. Each infant is represented by a consistent color across both panels.

**Table 1 nutrients-14-05224-t001:** Demographic data for the infants and their mothers (N = 8).

Variable	Mean (SD) or N (%)
Maternal age in years, mean (SD)	26.9 (3.9)
Maternal race (white)	7 (88%)
Antenatal steroids (yes)	7 (88%)
Mode of delivery (vaginal)	3 (38%)
Place of birth (inborn)	6 (75%)
Infant sex (female)	3 (38%)

**Table 2 nutrients-14-05224-t002:** Growth and nutrition parameters are provided as mean (SD) for the infants at birth, on the given study days, and at discharge (N = 8).

	Birth	Day 0	Day 7	Day 14	Day 21	Discharge
Postmenstrual age (weeks)	25.6 (1.4)	32.6 (2.7)	33.6 (2.7)	34.6 (2.7)	35.6 (2.7)	43.3 (2.5)
Feeding volume (mL/kg/day)			131 (18)	135 (13)	129 (14)	
Caloric intake (Kcal/kg/day)			113 (10)	119 (10)	115 (11)	
Weight (g)	816 (247)	1790 (778)	2044 (793)	2158 (843)	2433 (816)	3990 (666)
Weight (z-score)	0.05 (0.9)	−0.35 (0.5)	−0.17 (0.6)	−0.43 (0.6)	−0.28 (0.6)	−0.42 (0.35)
Head circumference (cm)	23 (1.9)	28 (2.6)	29 (2.5)	30 (2.4)	31 (2.4)	36 (1.2)
Head circumference (z-score)	−0.33 (0.7)	−0.73 (0.3)	−0.67 (0.4)	−0.65 (0.3)	−0.58 (0.6)	−0.55 (0.7)
Length (cm)	32 (2.9)	39 (4.1)	40 (3.6)	41 (3.9)	42 (3.8)	50 (1.6)
Length (z-score)	−0.40 (1.0)	−1.25 (0.6)	−1.19 (0.7)	−1.65 (0.6)	−1.70 (0.7)	−1.65 (0.9)

## Data Availability

Not applicable.

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
