# Peer review of "Postnatal Leptin Levels Correlate with Breast Milk Leptin Content in Infants Born before 32 Weeks Gestation"

_nutrients, 2022, doi:10.3390/nu14245224_

Round 1

Reviewer 1 Report

In this study, Chatmethakul and the colleagues reported postnatal serum leptin levels correlated with breast milk leptin content in infants. The authors investigated about 8 infants in cross-over trial. Although the number of study subjects was relatively small, the results were clear. However, there are some concerns remaining as followings.

Comments

1.     On would like to know the effects of leptin on the infants (such as metabolic factors (fasting plasma glucose, triglyceride, cholesterol, etc.), breast milk intake, body weight gain). Please demonstrate these data about leptin effects. Were there any significances between blood leptin concentration and these factors?

2.     One also would like to know the change in individual infant about leptin serum levels, blood metabolic factors, body weight gain, etc.) in this study. The study subject number was small, so the authors should demonstrate the precise individual data. These results would enhance readers’ understanding. 

Reviewer 2 Report

This is a very interesting study, however the study design needs some improvement. In this paper, the authors trying to conclude that leptin may play a role in the favorable growth profiles seen in formerly breast fed children, however there are many other hormones can play important roles during the early development of infants. It would be desirable to assay some of the other hormones, including cytokines and growth factors, that can affect early development in the maternal milk as well. 

1) Please measure some of the cytokines and growth factors in both the maternal breast milk and the infant plasma so that to understand the whole picture of changes in the infants that are under different feeding program.

2) In all of the figures, p values are presented, however what comparison was made was not very clear. Please label the comparative pair that the p value referred to. This refer to all of the comparisons and all the p values in the paper. 

Round 2

Reviewer 1 Report

The authors well-revised their manuscript. I have no further comments on this manuscript.